# Evaluation and Genetic Analysis of Parthenocarpic Germplasms in Cucumber

**DOI:** 10.3390/genes13020225

**Published:** 2022-01-25

**Authors:** Chenxing Gou, Pinyu Zhu, Yongjiao Meng, Fan Yang, Yan Xu, Pengfei Xia, Jinfeng Chen, Ji Li

**Affiliations:** 1State Key Laboratory of Crop Genetics and Germplasm Enhancement, College of Horticulture, Nanjing Agricultural University, No. 1 Weigang, Nanjing 210095, China; 2019104054@njau.edu.cn (C.G.); 2019204025@njau.edu.cn (P.Z.); 2018204015@njau.edu.cn (Y.M.); 2107204028@njau.edu.cn (F.Y.); 2020104057@stu.njau.edu.cn (Y.X.); jfchen@njau.edu.cn (J.C.); 2Nanjing Innovation Vegetable Molecular Breeding Research Institute, Nanjing 211899, China; njxczm@163.com

**Keywords:** cucumber, parthenocarpy, germplasm, evaluation, genetic analysis

## Abstract

Parthenocarpy is an important agronomic trait in cucumber (*Cucumis sativus* L.) production. However, the systematic identification of parthenocarpic germplasms from national gene banks for cucumber improvement remains an international challenge. In this study, 201 cucumber lines were investigated, including different ecotypes. The percentages of parthenocarpic fruit set (PFS) and parthenocarpic fruit expansion (PFE) were evaluated in three experiments. In natural populations, the PFS rates fit a normal distribution, while PFE rates showed a skewed distribution, suggesting that both PFS and PFE rates are typical quantitative traits. Genetic analysis showed that parthenocarpy in different ecotypes was inherited in a similar incompletely dominant manner. A total of 5324 single nucleotide polymorphisms (SNPs) associated with parthenocarpy were detected in a Genome-wide association study (GWAS) of parthenocarpy in the 31 cucumber lines, from which six parthenocarpic loci, including two novel loci (Pfs1.1 and Pfs4.1), were identified. Consequently, fifteen of the elite lines that were screened presented relatively stronger parthenocarpy ability (PFS > 90%, PFE > 50%), among which six cucumber lines (18007s, 18008s, 18022s, 18076s, 18099s, and 18127s) exhibited weak first-fruit inhibition. Three lines (18011s, 18018s, and 18019s) were screened for super ovary parthenocarpy, which showed more attractive performance. Four low-temperature-enhanced parthenocarpy lines (18018s, 18022s, 18029s, and 18012s) were identified, which were suited for breeding for counter-season production. Our approaches could help increase efficiency and lead to parthenocarpy improvements for modern cucumber cultivars.

## 1. Introduction

The initiation of fruit generally depends on successful pollination or fertilization. However, these processes are restricted by narrow environmental factors [1]. Cucumber (*Cucumis sativus* L.) is one of the most important fruit crops in the Cucurbitaceae family. As a kind of monoecious crop, a successful fruit set in cucumber depends on favorable conditions [2]. The yield of cucumber can be seriously reduced in the absence of pollinators or when planting under unsuitable environmental conditions, such as high humidity, weak light, and high temperature [3]. In 1902, Noll first introduced the term parthenocarpy in cucumbers to describe the formation of seedless fruit in the absence of functional pollination or other stimulation [4]. Breeders considered that parthenocarpy maybe provide an opportunity to overcome the problems of poor fruit setting caused by unfavorable pollinating conditions [5]. Moreover, parthenocarpic fruits tend to be firmer and fleshier than pollinated fruits owing to their lack of seeds [6]. However, only a few studies have so far attempted to evaluate and apply the parthenocarpic cucumber germplasm. In many cucumber production areas, the main cucumber cultivars possess relatively weak parthenocarpic ability, especially in China, where the yield of cucumber is limited (FAO, 2020).

Parthenocarpy is a complex trait in cucumber that is controlled by multiple phytohormones and genes. Parthenocarpy in cucumber can be artificially induced or manipulated via exogenous application of auxin, cytokinin, gibberellic acid, or brassinosteroids [7,8,9]. Yin et al. showed that overexpression of the auxin-synthesizing gene *DEFH9-IaaM* in cucumber could stimulate parthenocarpy [10]. In tomato, series of auxin-related genes were identified that played important roles involving the parthenocarpic fruit set [11,12,13,14]. Ren et al. reported that the overexpression of *SlTIR1* results in parthenocarpic fruit formation in tomato plants [15]. Our previous study also confirmed that the transcript levels of *CsTIR1* and *CsAFB2* decreased in parthenocarpic and pollinated fruits, suggesting that the decreased expression of these two genes would induce parthenocarpy fruit in cucumber [16,17]. Genetic studies on parthenocarpy in cucumber have been largely inconsistent for several decades, ranging from single gene to complex polygene inheritance [18,19,20,21]. Seven quantitative trait locus (QTLs) for parthenocarpy, including a major-effect QTL on chromosome 2, were detected in European greenhouse cucumber [22]. Meanwhile, the same numbers of parthenocarpy QTLs were identified in North American processing cucumbers, which showed incomplete overlap with the QTLs detected in European greenhouse genotypes [23]. Additionally, four novel parthenocarpy QTLs were identified from a South China ecotype cucumber [24]. Environmental factors such as temperature, photoperiod, light intensity, and nutritional conditions have considerable influences on parthenocarpy [11,25,26,27,28,29]. Many studies have reported that low temperature could induce parthenocarpy in different fruit crops such as tomato, eggplant, zucchini, melon, and cucumber [27,28,29]. In a previous study, the auxin content in the ovary increased under low night temperature, thereby inducing parthenocarpy in cucumber [30]. On the contrary, high temperature suppress parthenocarpy initiation by inhibiting the synthesis of auxin and gibberellin in the ovaries in cucumber [31]. Studies have suggested that short-daylight conditions could enhance parthenocarpy by increasing the activity of auxin [32]. Adequate nutrient supply is a prerequisite for fruit development. In other words, fruit development is dependent on sufficient nutrient availability, which also plays an essential role in parthenocarpy. The agricultural application of parthenocarpy was limited by environmental sensitivity and genotypes in a previous study [33]. To date, few studies have explored the genetic diversity of parthenocarpy and its relations to environmental factors in cucumber.

In this study, 201 cucumber germplasms were investigated, of which the parthenocarpy ability was identified and calculated in three independent experiments. The heritance of the strong parthenocarpy ability in cucumber was also analyzed. In addition, Genome-wide association study (GWAS) was conducted to detect novel genetic loci for parthenocarpy in cucumber based on genome resequencing data from 31 cucumber lines. These approaches would be helpful in better understanding the diversity of parthenocarpy in cucumber, to promote innovation regarding the parthenocarpic germplasm, and to increase the efficiency of parthenocarpy for modern cucumber cultivars.

## 2. Materials and Methods

### 2.1. Plant Materials and Grow Condition

In total, 201 cucumber lines were used in this study, including 19 European greenhouse ecotypes (EG), 27 U.S. processing ecotypes (UP), 80 South China ecotypes (SC), 68 North China ecotypes (NC), 6 Xishuangbanna ecotypes (XSBN), as well as a wild cultivar (*Var. hardwickii* Gabaev) preserved at the Laboratory of Cucurbit Genetics and Germplasm Enhancement, Nanjing Agricultural University. The non-parthenocarpy inbred line ‘18016s’ was used in all experiments as a reference non-parthenocarpy phenotype.

The parthenocarpy of 201 cucumber lines was evaluated in the autumn seasons of 2014, 2017, and 2018 in the Jiangpu Experimental Field of Nanjing Agricultural University, with 10 plants in each cucumber line. Normal seeds were sown in seedling-raising plates with media including a mixture of coco coir and vermiculite (v:v = 2:1) after germination. After sowing for 14 days, the seedlings were transplanted into plastic greenhouses. Plants grew on double-height ridges covered with plastic film (narrow spacing of 50 cm and plant spacing of 25 cm), with 25 cm spacing between plants. The field management was performed under normal cucumber production conditions. About 4 weeks after transplanting (one day before flowering), the female flowers were isolated with metal wires to avoid pollination in each plant.

### 2.2. Evaluation of Parthenocarpic Ability

The female flowers below the 5th node on the main stem were removed to reduce errors before parthenocarpy evaluation. The female flowers (6–30th nodes) that would open on the next day were all tied with metal wires to avoid pollination, and the colored labels were hung to mark the isolation time. The phenotypic data for each female flower were collected 8 days after the isolation treatment. Fruit samples over 4 cm in length and over 1 cm in diameter (the average length of a normal ovary is 3.50 cm and the diameter is 0.70 cm) were recorded and numbered as parthenocarpy. Meanwhile, parthenocarpic phenotypes, including expanded parthenocarpic fruit (Figure 1D), and parthenocarpic initial fruit, were recorded as parthenocarpic indices (Figure 1B). We then calculated the percentage of the parthenocarpy fruit set (PFS) [22] and the percentage of parthenocarpic fruit expansion (PFE). In addition, other traits, including the fruit weights of parthenocarpic and special parthenocarpic fruits, were indispensable to the evaluation of parthenocarpy in cucumber.

### 2.3. Low-Temperature Treatment

The low-temperature treatment was conducted in glass greenhouses at the Baima Teaching and Research Base of Nanjing Agricultural University in the winters of 2019 and 2020. In total, 20 plants from each cucumber line were planted in the greenhouses under 28 °C daily and 25 °C nightly temperatures during the seedling period. When the plants switched into full blossoming stage, half of each line (10 plants) was transferred into another low-temperature greenhouse with a 22 °C/15 °C (day/night) temperature regime. The evaluation of parthenocarpy was the same as above. Then, the fruit-related traits were compared, including PFS and the average fruit weight, length, and diameter for each cucumber line under the two temperature regimes.

### 2.4. Heredity of Parthenocarpy Traits

We also analyzed the phenotype data of the F1 genotype derived from crosses between 8 parthenocarpic accessions (18134s, 18021s, 18003s, 18093s, 18011s, 18014s, 18038s, and 18001s) and the non-parthenocarpic line 18026s.

In the genome-wide association study (GWAS) for 31 cucumber lines (Appendix A), DNA samples were extracted and purified from young leaves using the cetyltrimethylammonium bromide (CTAB) method as previously described by Murray and Thompson. The DNA concentration for each sample was adjusted to 50 ng/μL. All samples were sequenced on an Illumina HiSeq 2000 (Illumina Inc., San Diego, CA, USA). About 10 Gb of raw data for each sample was obtained. High-quality reads were retained by filtering low-quality data using the Trimmomatic package version 0.32 (Max Planck Institute of Molecular Plant Physiology, Brandenburg, Germany), while the sequencing depth and coverage for the 31 inbred cucumber lines are summarized in Appendix A. Filtered reads were mapped to the cucumber ‘9930’ reference genome (ftp://cucurbitgenomics.org/pub/cucurbit/genome/cucumber/Chinese_long/v3/, accessed on 17 September 2021) with Burrows–Wheeler alignment using the default parameters, and the result were exported as BAM format files using SAMtools. Single nucleotide polymorphism (SNP) calling was carried out using the HaplotypeCaller from the Genome Analyzer Tool Kit package version 4.2.0.0 (Broad Institute, Cambridge, UK) in GVCF mode (https://github.com/broadinstitute/gatk/releases, accessed on 20 September 2021), following best practice. The SNP dataset was filtered based on a 10% cutoff for missing data, and markers with minor allele frequencies ≥0.10 were considered for GWAS. The genome-wide association study (GWAS) was run using the fixed and random model of circulating probability unification (Farm CPU) implemented in R package genome association and prediction integrated tool (GAPIT, Washington State University, WA, USA) using the first PC calculated on genotypic data as a covariate [34,35]. Mark-trait associations (MTAs) are defined as SNPs, surpassing the significance threshold of a false discovery rate (FDR) < 0.05 according to Storey’s method [36]. Manhattan plots display a stringent Bonferroni threshold corresponding to a nominal *p*-value of 0.05 to aid the identification of the most significant SNPs [37].

### 2.5. Statistical Analysis

In this study, violin and box plots depicted the phenotypic data distribution of parthenocarpic PFS, PFE, and fruit weight by using the R package ggplot 2 (Rice University, Houston, USA). An analysis of variance (ANOVA) of the phenotype data was conducted with SPSS statistical analysis software (SPSS Statistics v22.0.0.0, IBM, USA), and significant differences were set at *p* < 0.05. The correlation of the PFE and fruit weight was calculated using the Spearman’s rank correlation coefficient. According to the description of cucumber parthenocarpy by Li et al. [38], parthenocarpy was divided into 2 grades based on PFE values, namely the low PFE group (PFE ≤ 40%) and high PFE group (PFE > 40%).

## 3. Results

### 3.1. Evaluation of Parthenocarpy in Cucumber Germplasm Resources

Wu et al. [39] and Clavin et al. [23] distinguished parthenocarpy phenotypes of cucumber into two types: initial parthenocarpic fruit and parthenocarpic fruit expansion (Figure 1). Referring to these studies, both the percentages of parthenocarpy fruit set (PFS) and parthenocarpic fruit expansion (PFE) for 201 cucumber lines were investigated to evaluate the parthenocarpy ability of cucumber germplasm resources. The phenotypic evaluation was performed in 2014, 2017, and 2018.

The distribution of the PFS across the 201 cucumber lines fit a normal distribution that ranged from 0% to 100%, suggesting that the PFS of cucumber is a quantitative trait. (Figure 2A). According to the PFS data, the cucumber germplasms were divided into 4 ranks, namely level I (0 ≤ PFS ≤ 25%), level II (25% < PFS ≤ 50%), level III (50% < PFS ≤ 75%), and level IV (75% < PFS ≤ 100%). In total, 37 cucumber lines were classed as level I (18.4%), of which 12 cucumber lines were identified as non-parthenocarpy accessions (PFS = 0.0%); 40 cucumber lines were classed as level II, 45 were classed as level III, and 79 were classed as level IV (Figure 2A; Appendix A). The PFS performances of the different ecological cucumbers were compared. The results showed that the European greenhouse (EG) cucumbers exhibited a generally higher parthenocarpy rate, of which 70% EG-type cucumber lines were classed as level IV (Figure 2B). In contrast, the XSBN-type cucumbers presented the worst parthenocarpy ability. The parthenocarpy rates of XSBN-type cucumbers were in the range of 12.5% to 37.5% (Figure 2B). We also classified the 201 cucumber lines based on phenotypic data for PFE using the same grading standard as for the PFS ranks. This PFE showed a similar skewed distribution. The PFE volumes from all lines were generally low, which for the majority of the germplasms were below 25% (Figure 2C). There were 120 accessions in level I (0 ≤ PFE ≤ 25%), 50 accessions in level II (25% < PFE ≤ 50%), 22 accessions in level III (50% < PFE ≤ 75%), and 9 accessions in level IV (75% < PFE ≤ 100%) (Figure 2C; Appendix A). Comparing the performances of different ecological cucumbers in terms of PFE, the results showed that the EG-type cucumber exhibited the highest PFE rates generally, most of which were higher than 37.5%. Additionally, the PFE rates of the NC-type cucumber were slightly lower than those of the EG-type, while the XSBN-type cucumber exhibited the worst parthenocarpy ability (Figure 2B,D). By joint analysis of PFS and PFE rates of the cucumber germplasms, only 15 cucumber lines with high PFS and PFE rates (PFS > 90%, PFE > 50%) were selected, which were considered as high-value breeding lines for parthenocarpy improvements in cucumber (Figure 2D, Appendix A).

### 3.2. Effect of First-Fruit Inhibition in Parthenocarpy Cucumbers

In cucumber, a phenomenon known as “first-fruit inhibition” (FFI) has been well studied, whereby the nutritional requirements of preexisting fruit usually inhibit the development of the subsequent female flower maturation and fruit set. In cucumber production, this limited simultaneous fruit set pattern greatly reduces the yield, especially in once-over harvest systems [40]. Mitra et al. found that fertilized ovaries could be recovered after FFI removal but parthenocarpic fruit could not [41]. In this study, the effect of first-fruit inhibition in the above 15 screened parthenocarpy cucumber lines was investigated by maintaining all of the parthenocarpic fruit on the plants. High parthenocarpy fruit expansion rates (PFE > 80.0%) were identified in six parthenocarpic lines, including 18007s, 18008s, 18022s, 18076s, 18099s, and 18127s, suggesting that these cucumber lines presented suppressed first-fruit inhibition (Figure 3A) that caused high parthenocarpy fruit expansion rates (PFE > 80.0%). The correlation analysis of the single fruit weight and PFE rates of different ecotypes found that PFE rates were negatively correlated with single fruit weight (r < 0). This negative correlation was most significant in the EG cucumber type (|r| = 0.636). The accessions with lower PFE rates produced significant larger fruit than the low-PFE lines (Appendix A).

### 3.3. Screening of Super Ovary Parthenocarpic Germplasms

The “super ovary” is a special type of parthenocarpy in cucumber that refers to an ovary that is larger than normal at anthesis [26,32,42]. There was no significant difference in fruit length between the two types of ovaries. The commercially mature fruit from super ovary cucumbers are straight and have flowers remaining on the tip, which are regarded as more attractive and preferred in the fresh market. In this study, three cucumber lines (18011s, 18018s, and 18019s) were identified that produced super ovaries with 4~5 d delay of corolla opening, including one SC-type and two NC-type cucumbers (Figure 3B). To date, no study has identified the super ovary germplasm in other cucumber ecotypes. Hence, these germplasms provide the basis for studying the genetic characteristics and formation mechanism of super ovaries.

### 3.4. Exploring Low-Temperature Enhanced Parthenocarpic Germplasms

As a subtropical originated crop, cucumber is particularly sensitive to low-temperature, which can suppress not only the vegetative growth of the plant, but also the development of flowers and fruit. Interestingly, low-temperature can induce parthenocarpy in some cucumber genotypes. This sort of low-temperature-activated parthenocarpy was also detected in tomatoes, eggplants, zucchinis, and melons [26,27,28,29]. Low-temperature-tolerant or enhanced-parthenocarpy germplasms would be ideally suited for counter-season production, e.g., in winter and early spring. To explore low-temperature-activated parthenocarpic germplasms, 201 cucumber lines were cultivated in low-temperature conditions (day/night = 22/15 °C and 16 h/8 h) and normal temperature conditions (day/night = 28/25 °C and 16 h/8 h) separately. The PFS and parthenocarpic fruit weight, diameter, and length values of these cucumber lines were compared between the different growth conditions. The results showed that only four cucumber lines (18018s, 18022s, 18029s, and 18012s) displayed significantly increasing rates of parthenocarpic fruit set under lower temperature conditions (Figure 4A). However, the size of parthenocarpic fruit in 18029s plants was significantly decreased in low-temperature conditions (Figure 4B–D). Surprisingly, both the PFS (60%) and PFE (60%) rates of 18012s plants, recognized as a non-parthenocarpy line by field experiments, were elevated to level IV by the low-temperature treatment. Transcriptome and physiological analyses suggested that the low-temperature-induced parthenocarpy in 18012s was regulated by the crosstalk of auxin and ethylene [43].

### 3.5. Inheritance Analysis of Parthenocarpy

The normal distributions of PFS and PFE in 201 cucumber lines indicated that parthenocarpy in natural germplasms was a quantitative trait. Sixteen hybrids crossed by eight parthenocarpy accessions and one non-parthenocarpy line were generated to investigate the inheritance of parthenocarpy in cucumber. All F1 progeny exhibited parthenocarpy ability, although the PFS rates of these hybrids (ranging from 37.1% to 47.7%) were relatively lower than their donor parents (Table 1). These results confirmed that parthenocarpy was inherited in an incompletely dominant manner and controlled by multiple loci and genes.

### 3.6. Identification of SNPs Associated with Parthenocarpy through GWAS

Genome-wide association studies (GWAS) have been widely used to identify genetic variants affecting complex traits, either by comparative analysis or correlation analysis, and have identified many SNPs associated with target traits. In this study, we performed a GWAS of parthenocarpy of the 31 accessions using a compressed general linear model (GLM). A total of 5324 jointly associated SNPs for PFE were detected that consistently exceeded a significant threshold of −log 10 (P) ≥ 3.0. For the convenience of presentation and the summary of results, the chromosomal region at which adjacent pairs of associated SNPs were less than 1 Mb distant was defined as a single locus. The six total loci consisted of two novel parthenocarpic loci (Pfs1.1, Pfs4.1) and four reported loci (Pfs1.1, Pfs3.1, Pfs3.2, Pfs4.1, Pfs6.1, Pfs6.2), while 1829 associated SNPs were detected (Figure 5), of which Pfs3.1 and Pfs3.2 were overlapped with the reported QTLs Parth3.1 and Parth3.2 [26]; and Pfs6.1 and Pfs6.2 were overlapped with the reported QTL Parth6.2 [11]. A total of 129 nonsynonymous SNP variations were identified that were located in the exonic regions of 80 genes, of which an auxin-responsive gene *CsaV3 6G039630* was screened, which was annotated as a homolog of ARF17 in arabidopsis and may have a function in cell growth (Appendix A).

## 4. Discussion

### 4.1. Abundant Diversity of Parthenocarpy in Cucumber

The evaluation and identification of valuable parthenocarpy germplasms is of great significance for crop improvements. Naturally occurring (genetic) parthenocarpy has been observed in many plants. Previously, 113 citrus accessions, including pummelo, mandarin, and their relatives, were investigated, 63 of which showed autonomous parthenocarpy and indicated that ‘Kunenbo’ has a very high degree of autonomous parthenocarpy [44]. The parthenocarpy ability of 113 eggplant germplasm resources was evaluated by stigma excision. D-11 with complete parthenocarpy was identified from 4 parthenocarpy accessions [45]. Yoshioka et al. explored 172 accessions from an East Asian melon collection and identified accessions. They also compared the emasculation and stigma excision methods, finding that stigma excision is superior to emasculation for the evaluation of parthenocarpic ability [46]. De Ponti suggested that the percentage of the parthenocarpy fruit set (PFS) was an important indicator for evaluating parthenocarpy ability [47].

As the first recorded parthenocarpy crop, cucumber exhibits more abundant diversity of parthenocarpy by contrast to citrus, grape, and solanaceous crops [7,48,49,50]. Although in different cucumber ecotypes parthenocarpy was inherited in a similar incompletely dominant manner, various parthenocarpy performances were observed and studied in this study. For instance, super ovary parthenocarpy was first defined by Rudich et al. [32] based on the phenotype of a parthenocarpic monoecious cucumber line [26,42,51]. A super ovary refers to an ovary that is larger than normal at anthesis. Sun et al. suggested that the main difference in the formation of normal and super ovaries is the delayed corolla growth and opening of the super ovary [52]. In present study, three cucumber accessions were identified that presented super ovary parthenocarpy, which provided an opportunity to uncover the genetic and molecular basis of super ovary parthenocarpy. Studies on the influence of temperature on parthenocarpy in *Cucumis sativus* L. are unanimous in their conclusion that lower temperatures stimulate parthenocarpy. However, heritable low-temperature-induced parthenocarpy has not been reported yet in cucumber. Four heritable low-temperature-induced parthenocarpy cucumber lines were screened in this study. The parthenocarpy performance of 18012s was extremely sensitive to low temperature and was considered to be an excellent material for revealing the role of temperature during the fruit set.

### 4.2. Comparison of Parthenocarpic Features between Different Cucumber Ecotypes

Cucumber is a widely cultivated vegetable globally. According to the plant characteristics and main planting distribution areas, cucumber varieties are divided into five ecological types, which can be directly distinguished according to the characteristics of the fruit. The fruit of European greenhouse and U.S. processing cucumbers are relatively small, while the surfaces of the EG fruit are smooth and UP cucumber fruit exhibit abundant fruit tumors. Meanwhile, the fruit of Asian-type cucumbers are relatively large. Most of North China ecotype cucumber cultivars produce very long fruit with obvious fruit stalks and a tapering fruit apex, whereas the majority of South China and Xishuangbanna ecotype cucumbers produce thick and round fruit. In the present study, distinctive parthenocarpic characteristics were also identified in different cucumber ecotypes. For instance, the parthenocarpic fruit set rates of EG, UP, and NC cucumbers were relatively higher than for the SC cucumbers, while XSBN cucumbers lacked parthenocarpy ability (Figure 2B). Asian-type cucumbers showed more abundant parthenocarpic diversity, e.g., super ovary parthenocarpy was only detected in SC and NC ecotypes (Figure 3B) [52,53].

Interestingly, the “tolerance” of first-fruit inhibition was only observed in EG- and UP-type cucumbers (Figure 3A), while regarding PFE rates, 46% of NC-type and 66% SC-type cucumbers were below 25% (Figure 2D; Appendix A). Many studies suggested that the inhibition of parthenocarpy fruit expansion caused by the first-fruit inhibition, which should not be ignored during the evaluation of parthenocarpy germplasms [23,54,55,56]. In recent decades, a series of EG and UP type parthenocarpic germplasms without FFI effects have been identified [57,58], and the parthenocarpy ability of major varieties in American and European countries have been improved using these resources. In contrast, the parthenocarpy ability of SC and NC cultivars is generally weak due to the serious FFI effects on the NC and SC parthenocarpic germplasms. In large parts of Chinese cucumber production areas, exogenous hormone treatment is widely used to maintain fruit expansion rates. It is widely considered that the inhibitory effect of the first fruit is caused by nutrient competition between fruit. The nutritional requirements of preexisting fruit usually inhibit the development of the subsequent female flower maturation and fruit set. The correlation analysis showed that there was a negative correlation between fruit weight and PFE across all ecotypes (Appendix A). We speculated that there may be a significant correlation between first-fruit inhibition and fruit weight in cucumber.

### 4.3. Complex Genetic Basis of Parthenocarpy in Cucumber

Genetic studies have been largely inconsistent regarding the mode of inheritance for parthenocarpy in cucumber, and have ranged from proposals of a single gene to complex multigenic inheritance [18,19,20,21,59,60,61,62]. Sun et al. reported 10 QTLs associated with parthenocarpy and observed significant epistasis and large genotype × environment interactions for this trait in the North American processing cucumber [62]. Recently, based on a novel phenotypic approach, the parthenocarpy QTLs in North American processing cucumber were re-examined. Seven QTLs associated with parthenocarpic fruit set were detected, which were located on chromosomes 2, 4, 5, 6, and 7. Among them, parth5.1, parth6.1, parth6.2, and parth7.1 were consistently identified in all experiments [23]. In our previous study, seven QTLs for parthenocarpic fruit set were detected in a European greenhouse cucumber with a major-effect QTL parth2.1 in chromosome 2 [39]. Although distinctive QTLs have been identified in different cucumber ecotypes, they could not explain the complete variation in parthenocarpy. GWAS is an advanced strategy that provides a systematic and relatively unbiased screening of the whole cucumber genome, which can lead to the discovery of unsuspected susceptibility variants. The population size available for GWAS analysis has a great impact on the statistical power in detecting associations between DNA variants and traits. The use of a small population for GWAS may increase the false positives of SNPs associated with traits [63,64].

In the present study, a total of six loci associated with parthenocarpy were detected by GWAS analysis of 31 inbred cucumber lines. The detected loci Pfs3.1 and Pfs3.2 in this study were overlapped with the reported QTLs Parth3.1 and Parth3.2 [26], while the detected loci Pfs6.1 and Pfs6.2 in this study were overlapped with the reported QTL Parth6.2 [11], suggesting that the results detected by GWAS based on the small population size may be consistent with a previous study of QTL mapping for parthenocarpy. However, we will verify the accuracy of these loci and SNPs via repeated GWAS analyses with larger samples and combined with QTL mapping in the future.

## 5. Conclusions

The studies on parthenocarpic cucumbers focus on EG- and NC-type cucumbers, although more abundant parthenocarpic resources have been identified, while there are few excellent parthenocarpic resources in other ecotypes. In this study, the percentages of parthenocarpy fruit set and parthenocarpic fruit expansion in 201 inbred cucumber lines were investigated to evaluate the cucumber germplasm resources, and 15 accessions with strong parthenocarpy ability were screened, which can be used to breed new varieties with stable parthenocarpy. In order to study the molecular mechanism of parthenocarpy and for the breeding of new varieties, it is necessary to enrich the channels of parthenocarpic materials by using molecular biotechnology, including EMS chemical mutagenesis, overexpression or gene silencing, the CRISP/Cas9 approach, as well as TILLING and other techniques, on the basis of extensive existing collections of cucumber parthenocarpy germplasm resources in order to create new parthenocarpic resources.

## Figures and Tables

**Figure 1 genes-13-00225-f001:**
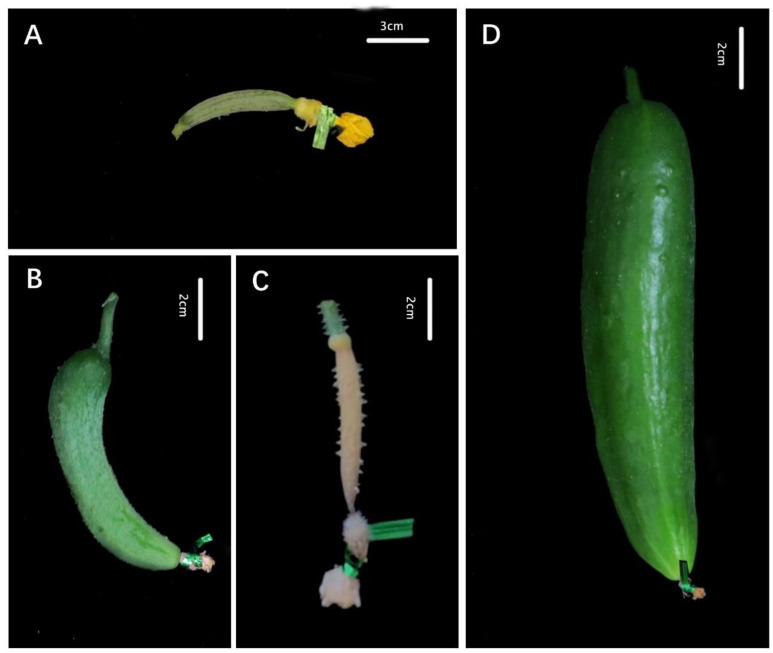
Developmental stages of parthenocarpy and non-parthenocarpy in cucumber at 8 days after the isolation treatment: (**A**) anthesis day; (**B**) initial parthenocarpic fruit set that later stopped; (**C**) aborted fruit; (**D**) expanded parthenocarpic fruit. Expanded parthenocarpic fruit: isolated ovary can normally develop into fruit. Parthenocarpic initial fruit: isolated ovary stops growing or has grown slightly and become aborted; this fruit can continue to develop into normal fruit after external stimulation. Percentage of parthenocarpy fruit set (PFS) = number of parthenocarpic fruit/number of ovaries with isolated treatment × 100%. Percentage of parthenocarpic fruit expansion (PFE) = number of expanded parthenocarpic fruit/number of ovaries with isolated treatment × 100%.

**Figure 2 genes-13-00225-f002:**
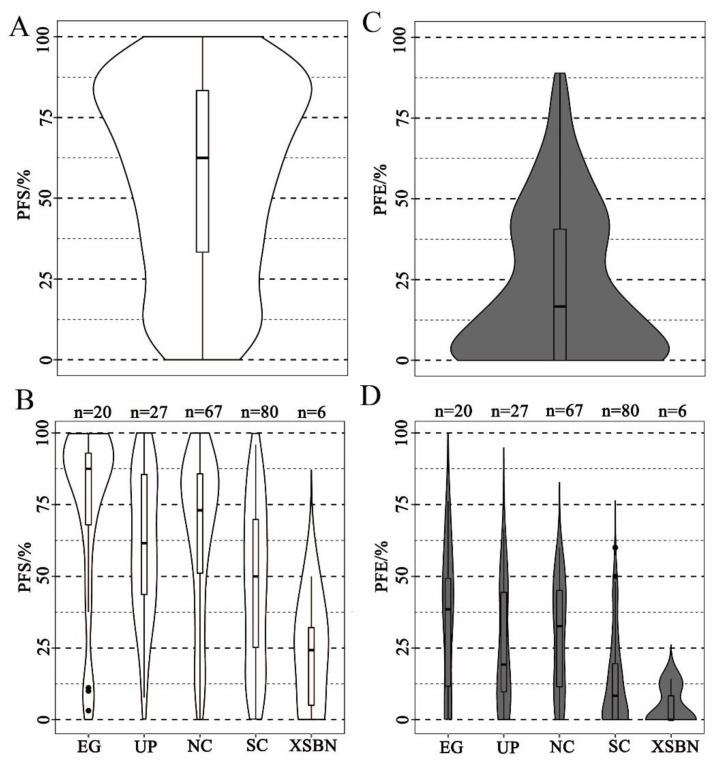
Violin and box plots depicting phenotypic distributions of PFS and PFE: (**A**,**C**) phenotypic distributions of PFS and PFE across 201 cucumber lines; (**B**,**D**) phenotypic distributions of PFS and PFE across different ecotypes. EG: European greenhouse ecotype; UP: U.S. processing ecotype; NC: North China ecotype; SC: South China ecotype; XSBN: Xishuangbanna ecotype. Phenotype data were based on the mean phenotype data for 2014F, 2016F, and 2017F. Dark spot indicates discrete values of inbred cucumber lines in box plots, which are indicate that the observed values are more than three times the quartile range from the median value.

**Figure 3 genes-13-00225-f003:**
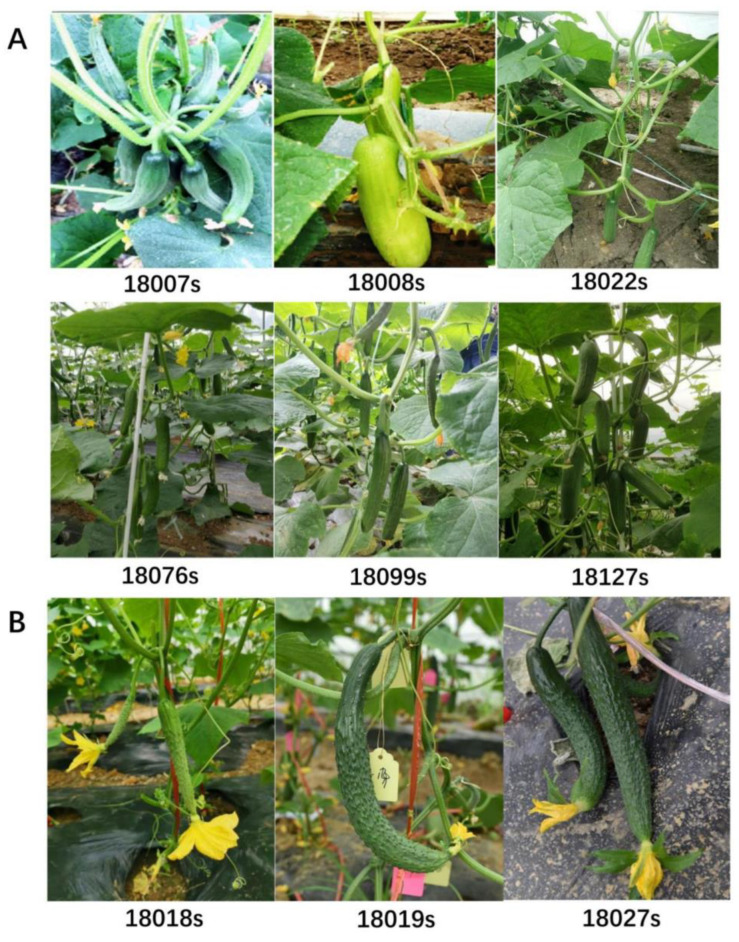
Performance of strong parthenocarpy without first-fruit inhibition and super ovary parthenocarpy: (**A**) phenotypes of parthenocarpy without first-fruit inhibition (multiple parthenocarpic fruits without first-fruit inhibition (18007s, EG); parthenocarpic fruiting on main stem without first-fruit inhibition (18008s, EG; 18022s, UP; 18076s, EG; 18099s, EG; 18127s, UP); (**B**) phenotypes of super ovary parthenocarpy in cucumber (18018s, SC; 18019s, NC; 18027s, NC).

**Figure 4 genes-13-00225-f004:**
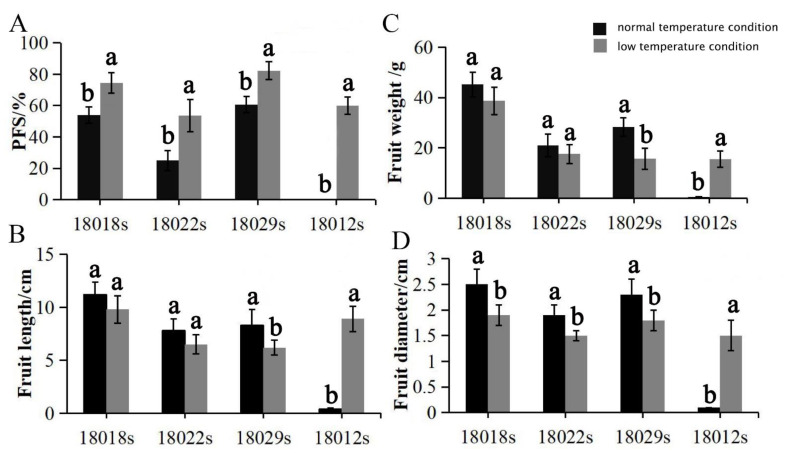
Effects of two temperature conditions on the PFS, fruit weight, fruit length, and fruit diameter values: (**A**) effects of the two temperature conditions on PFS rates; (**B**) effects of the two temperature conditions on parthenocarpic fruit length values; (**C**) effects of the two temperature conditions on parthenocarpic fruit weight values; (**D**) effects of the two temperature conditions on parthenocarpic fruit diameter values. Note: Letters a,b followed by the same letter are not significantly different at the α = 0.05 level.

**Figure 5 genes-13-00225-f005:**
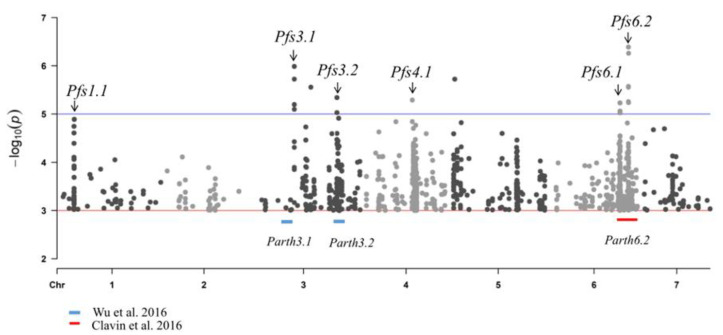
Manhattan plot for parthenocarpy based on 31 re-sequence dates of cucumber lines. The plot shows individual SNPs across all chromosomes (*x*-axis) and −log 10 (*p*) values of each SNP association (*y*-axis). Six putative loci (Pfs1.1: Chr1,3326820–3695442, 368.6 kb, consisting of 308 SNPs; Pfs3.1: Chr3, 15857769-16956515, 1098.7 kb, consisting of 12 SNPs; Pfs3.2: Chr3, 29681685–30523468, 841.8 kb, consisting of 428 SNPs; Pfs4.1: Chr4, 15377879–15897460,519.6 kb, consisting of 346 SNPs; Pfs6.1: Chr6, 23211119–24135690,925.7 kb, consisting of 339 SNPs; Pfs6.2: Chr6, 26053985–26455152,401.2 kb, consisting of 396 SNPs) are marked by black arrow heads above the blue line. Parth3.1 and Parth3.2 on chromosome 3 (marked by blue lines) and parth6.2 on chromosome 6 (marked by red lines) were previously identified by Wu et al. and Clavin et al. [23,39].

**Table 1 genes-13-00225-t001:** Frequency of parthenocarpic and non-parthenocarpic F1 plants derived from a cross between parthenocarpic cucumber lines and a non-parthenocarpic cucumber line 18016s.

Cucumber Lines	No. Plants	No. Isolated Ovaries ^a^	No. Parthenocarpic Plants ^b^	No. Parthenocarpic Fruit ^c^	Parthenocarpy Fruit Set (%)
Parental lines					
18016s	10	90	0	0	0
18134s	10	90	10	85	94.4
18021s	10	90	10	84	93.3
18003s	10	90	10	86	95.5
18093s	10	90	10	84	93.3
18011s	10	90	10	83	92.2
18014s	10	90	10	79	87.7
18038s	10	90	10	82	91.1
18001s	10	90	10	74	82.2
F1 progeny					
18134s × 18016s	10	90	10	43	47.7
18016s × 18134s	10	90	10	38	42.2
18021s × 18016s	10	90	10	36	40
18016s × 18021s	10	90	10	32	35.5
18003s × 18016s	10	90	10	28	31.1
18016s × 18003s	10	90	10	29	32.2
18093s × 18016s	10	90	10	37	41.1
18016s × 18093s	10	90	10	31	34.4
18011s × 18016s	10	90	10	32	35.5
18016s × 18011s	10	90	10	37	41.1
18014s × 18016s	10	90	10	33	36.6
18016s × 18014s	10	90	10	37	41.1
18038s × 18016s	10	90	10	42	46.6
18016s × 18038s	10	90	10	35	38.8
18001s × 18016s	10	90	10	37	41.1
18016s × 18001s	10	90	10	35	38.8

**^a^** Number of ovaries isolated with metal wire. **^b^** total number of plants with at least two parthenocarpic fruit samples without rotten parts at 8 days after isolation treatment. **^c^** total number of parthenocarpic samples fruit without rotten parts at 8 days after isolation treatment.

## Data Availability

Not applicable.

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
