# Peer review of "Evaluation and Genetic Analysis of Parthenocarpic Germplasms in Cucumber"

_genes, 2022, doi:10.3390/genes13020225_

Round 1
Reviewer 1 Report
Restructure the abstract and add more information.
The introduction lacks a clear hypothesis.
Results are not comprehensively written and can be elaborated.
Discussion can be improved from examples for the literature and more references to relate the results obtained.
Also, update and replace old references with recent references of the introduction section.
Add an appropriate conclusion; this section looks more like a results.
Author Response
Dear editor and reviewer,
We have studied comments carefully and have made corrections. We uploaded the revised manuscript, figures and tables. Please see the attachment.

Reviewer 2 Report
Accept
Author Response

(The authors gave the same response as above.)

Reviewer 3 Report
In this manuscript, Gou et al. reported a comprehensive genetic analysis of Parthenocarpic germplasms in cucumbera.
Major:
I am concerned about GWAS part. The authors need to address the impact of small sample size. How confident about the SNP quality and GWAS results.
The method description needs more clarity. Only mentioning BWA aligner is not enough. How do the raw reads align to the genome and how SNPs are called? What is the coverage of sequencing? The SNP calling tool mentioned is ‘Affymetirx ® Power Tools software package v1.18’, to my knowledge it is an array based solution only for proprietary GeneChip™ and Axiom arrays from ThermoFisher. There is no mentioning of using any array-based SNP detection in the text. Does ThermoFisher have cucumber SNP array, or is the array used custom-built? The lack of clarity undermines the confidence of the reported results.
Minor:
1. Fig. 2. Need some improvement.
a) Remove the top white space of panels B and D.
b) Where there are only three dots on EG of panel B and two on SC of panel D.
Please add all sample points to the violin plot and provide statistical comparisons results among groups. Please at least put the sample size of each group in the figure legend if not onto the figure panels.
2. Fig. 3. Please provide sample group info.
Author Response

(The authors gave the same response as above.)

Round 2
Reviewer 1 Report
The authors have answered all my previous questions, and I was pleased to see that the manuscript was significantly improved.
This manuscript is a resubmission of an earlier submission. The following is a list of the peer review reports and author responses from that submission.